# Lipotoxic Impairment of Mitochondrial Function in β-Cells: A Review

**DOI:** 10.3390/antiox10020293

**Published:** 2021-02-15

**Authors:** Axel Römer, Thomas Linn, Sebastian F. Petry

**Affiliations:** Clinical Research Unit, Center of Internal Medicine, Justus Liebig University, 35392 Giessen, Germany; Axel.Roemer@ernaehrung.uni-giessen.de (A.R.); Thomas.Linn@innere.med.uni-giessen.de (T.L.)

**Keywords:** lipotoxicity, free fatty acids, oxidative stress, mitochondrial dysfunction, beta cell, diabetes mellitus, polyphenol, ageing

## Abstract

Lipotoxicity is a major contributor to type 2 diabetes mainly promoting mitochondrial dysfunction. Lipotoxic stress is mediated by elevated levels of free fatty acids through various mechanisms and pathways. Impaired peroxisome proliferator-activated receptor (PPAR) signaling, enhanced oxidative stress levels, and uncoupling of the respiratory chain result in ATP deficiency, while β-cell viability can be severely impaired by lipotoxic modulation of PI3K/Akt and mitogen-activated protein kinase (MAPK)/extracellular-signal-regulated kinase (ERK) pathways. However, fatty acids are physiologically required for an unimpaired β-cell function. Thus, preparation, concentration, and treatment duration determine whether the outcome is beneficial or detrimental when fatty acids are employed in experimental setups. Further, ageing is a crucial contributor to β-cell decay. Cellular senescence is connected to loss of function in β-cells and can further be promoted by lipotoxicity. The potential benefit of nutrients has been broadly investigated, and particularly polyphenols were shown to be protective against both lipotoxicity and cellular senescence, maintaining the physiology of β-cells. Positive effects on blood glucose regulation, mitigation of oxidative stress by radical scavenging properties or regulation of antioxidative enzymes, and modulation of apoptotic factors were reported. This review summarizes the significance of lipotoxicity and cellular senescence for mitochondrial dysfunction in the pancreatic β-cell and outlines potential beneficial effects of plant-based nutrients by the example of polyphenols.

## 1. Introduction

The onset and progression of diabetes mellitus (DM) is crucially determined by the deterioration of the glucose-stimulated insulin secretion (GSIS) of pancreatic β-cells. In 2019, there were more than 460 million patients with DM worldwide with a steadily rising prevalence (9.3%) over the last few decades [1]. The impaired action of insulin in these patients leads to elevated plasma glucose levels. Chronic hyperglycemia can damage various molecules and tissues by glycation [2]. To prevent these complications, the supply with an adequate amount of insulin is necessary. As insulin secretion demands a lot of biochemical energy [3] and mitochondria contribute to 98% of cellular adenosine triphosphate (ATP) [4], their proper function becomes a major aspect in developing β-cell dysfunction and decay.

The progression of type 2 diabetes mellitus (T2DM) is accompanied by elevated free fatty acids (FFAs) [5,6,7] as well as the deterioration of the lipid metabolism. FFA are known to impair the function of β-cells and promote their failure by various mechanisms [8], among others, by toxic metabolites of lipid degradation, the activation or dysregulation of signaling pathways, oxidative stress, and an altered energy production. By mediating this so-called lipotoxicity, FFA can impair the mitochondrial metabolism and other compartments of the β-cell and disturb its capacity of both insulin synthesis and release. FFAs increase insulin resistance by several signaling pathways, including altered translocation of glucose transporter (GLUT). FFAs are therefore one of the major promotors of developing T2DM [9]. Nevertheless, FFAs are physiologically required for energy demands, as components of membranes or signaling molecules, and there is no defined qualitative or quantitative cut-off at which toxic effects commence. In general, lipotoxicity is defined as the impairment of cellular functions like mitochondrial respiration, protein translation and function, and induction of cell death mediated by the accumulation of FFAs. However, the underlying mechanisms are more complex and involve an imbalance between uptake, storage, and utilization of FFAs.

Beside elevation of FFA blood and tissue levels, and associated disorders of lipid metabolism, cellular senescence [10] is the second major contributor to the increasing prevalence of T2DM [11]. The rate of apoptotic events is increased while proliferation is disfavored with age, leading to an incremented decay of the endocrine pancreas. Increased levels of reactive oxygen species (ROS) and DNA damage diminish the regenerative capacity and function of β-cells correlating with a general impairment of insulin production and secretion machinery as well as an increased rate of apoptosis. There is also a connection between enhanced cellular stress leading to an age-dependent impairment of the lipid metabolism marked by increased plasma triglyceride (TG) levels and reduced postprandial TG clearance rates [12]. Likewise, there is a correlation between increased plasma FFA and age [13]. Concomitant with a decreased antioxidative capacity of β-cells [14], which is yet challenged by both FFA and age [15], this would promote lipotoxic effects ultimately leading to an acceleration of senescence and dysfunction of β-cells.

There are available data which point towards beneficial effects of plant-based nutrients [16]. Their beneficial effects are mainly, but not completely, thought to be mediated by their phytochemicals, consisting of heterogeneous substances with a variety of different bioactive molecules [17]. Beside e.g., carotenoids, glucosinolates, lectins, terpenes, alkaloids, and polysaccharides [18,19], the vast group of polyphenols has been investigated extensively for positive effects to improve insulin resistance and blood glucose levels [20]. Polyphenols can contain molecules like tannins, flavonoids, anthocyanins, proanthocyanidins, or derivatives of different organic acids [21,22,23], which are able to enhance action of insulin [24], glucose transport [25], or decrease intestinal carbohydrate hydrolysis [26,27,28]. These compounds can modulate antioxidative enzymes or cellular stress responses by gene expression [29,30,31] or regulation of cytokines and signaling pathways to improve β-cell function. By scavenging radicals and inducing the upregulation of antioxidative enzymes, phytochemicals can mitigate oxidative stress [32]. They directly control single steps in the lipid metabolism like uptake and storage of FFA to abate their toxic intermediates [33]. Moreover, phytochemicals can directly reverse the lipotoxic effects of FFAs by counteracting their adverse regulatory effects, e.g., by downregulating the respective signal pathways of insulin secretion or apoptosis [34]. There is a growing body of evidence that polyphenols could be able to reverse the negative effects of lipotoxicity and preserve, restore, and promote the physiological functions of β-cells. 

The aim of this review is to (I) summarize the knowledge of basic research on lipotoxicity directed against β-cells, in particular their mitochondria, with special regard to methodology, (II) elucidate the connection with cellular senescence, and (III) outline potential beneficial effects of dietary measures employing polyphenols as an example. 

## 2. Literature Search

The literature search based for this review was executed on 7th of September and 31st of December 2020 on PubMed. The search term (lipotox * OR “free fatty acid *”) AND (mitochondria * OR polyphenol * OR flavonoid * OR ageing) AND “beta cell *” yielded a total of 149 primary articles. The publication dates ranged from February 1977 to September 2020. After screening literature, 22 articles were excluded. Exclusion criteria were no suitable topic (11 exclusions), not written in English (five exclusions), reviews with no primary data (five exclusions) and no available full-texts (one exclusion), leaving 127 articles. Three reviews with primary data have been included. A flow chart of the literature search is given in Figure 1. A list of all screened articles is provided in Appendix A.

## 3. Factors Influencing Lipotoxic Outcomes in Tissue Culture

Since lipotoxicity cannot merely be defined by exceeding a specified concentration of FFA, the respective experimental setup must be considered closely when assessing experimental outcomes. 

The employed amount of FFA is one of the key factors for activating lipotoxic pathways. However, only few authors state the rationale for the chosen concentrations. A point of reference could be the serum FFA level of healthy and diabetic subjects and animals.

Spectroscopic analysis of blood samples revealed that the total FFA concentration in diabetic human serum is ranging from 3.5 to 15 mM, also giving concentrations of specific FFA like oleic acid (OA) with 0.74–3.9 mM and palmitic acid (PA) with 1.0–3.8 mM [35,36,37,38]. These data also reveal an increase of OA and PA concentrations in diabetic compared to healthy individuals of approximately 10% [36]. Serum concentrations of rodents are lower with a total FFA serum concentration of 0.8–1.5 mM [39,40]. Authors claim that a FFA concentration range of 0.5–2.0 mM is suitable to mimic lipotoxicity in T2DM [41,42]. There is no detailed information about specific FFA concentrations for pancreatic tissue, which would indicate suitable concentrations for experimental models. Given the distinct magnitude of β-cell dysfunction on parameters like ATP production or insulin secretion, the increase of 10% in FFA concentration appears to be insignificant. These facts could indicate that considering the total FFA concentration is not a satisfactory parameter to determine a lipotoxic environment. A small increase of FFA concentrations could be sufficient to have greater impact on deteriorations of β-cell metabolism. All these facts are raising the issue which concentrations should be employed for inducing lipotoxicity in an experimental setup. The screening of the available literature revealed significant differences. Most commonly 500 µM were used with a total range of 10–2000 µM. From the total of performed treatments with FFA (*n* = 104), there are available data for the following concentration ranges: 10–99 µM (*n* = 12), 100–499 µM (*n* = 53), 500 µM (*n* = 49), 501–2000 µM (*n* = 18) (Table 1). Single (*n* = 82) as well as multiple concentrations (*n* = 22) have been employed.

Insulin secretion and ATP levels were frequently examined. Treatment with 400 or 500 µM FFA led to a 6–90% decrease in ATP [4,43,44,45]. An incremented concentration of 2000 µM did not mediate higher toxicity [46]. A 15–70% diminished insulin secretion was induced by a short-or medium-term treatment (up to 72 h) with 400 or 500 µM FFA [47,48]. Exposure to 100 µM for several weeks led to a 20% decrease [49]. Glucose stimulation treatment was conducted with comparable concentrations of around 25 mM. The available data did neither reveal a dose dependent effect of FFA on insulin secretion, nor on ATP levels. Since the described protocols differed a lot in detail, e.g., regarding solvents, bovine serum albumin (BSA) amount, and incubation times, results were only comparable to a limited extent. For other readouts, the number of studies was too small to examine dose dependency.

In most tissue culture reports single rather than multiple FFA were examined. Hence, another key modulator for in vitro tissue culture is the selected type of FFA. It is noteworthy that only a small number of FFA have been used repeatedly for studying molecular pathways of cells. The most common design was the usage of single PA (*n* = 52), followed by the combination of PA + OA (*n* = 20), the combination of PA + OA + any additional FFA (*n* = 8), the combination of PA + any additional FFA (*n* = 6), treatment with single OA (*n* = 6), or with any single FFA (*n* = 4). Concentration range of combined FFA culture was 100–2000 µM (PA + OA), 200–1000 µM (PA + OA + any FFA) and 10–500 µM (PA + any FFA). The group of FFA used occasionally includes myristic, stearic, palmitoleic, linoleic, linolenic, methylpalmitic, docosahexaenoic, and arachidonic acid (Table 2). The combination of different types of FFA could be a suitable model for mimicking a physiological FFA composition [50].

Furthermore, the preparation of FFA solutions can have a major impact on the outcome depending on the applied assays. For dissolving FFA, the reviewed articles used ethanol (*n* = 25), NaOH or NaCl (*n* = 12), dimethyl sulfoxide (*n* = 4), and methanol (*n* = 3), while most articles did not specify how the FFA solutions have been prepared (*n* = 54) (Table 3). The use of solvents should be tested on key parameters like viability and also insulin concentration because they can affect the outcome by varying cytotoxic properties [51,52]. In addition, the total BSA concentration as well as the molar FFA:BSA ratio can have a drastic impact on the mediated lipotoxicity as well as specifically on the MTT viability assay [53,54], and were only described in detail in half of the studies (*n* = 53). The utilized BSA content in the respective cell culture media varied markedly, ranging between 0.05–5% for comparable FFA concentrations [55,56]. The molar FFA:BSA ratio determines the amount of unbound FFA in the treatment media, representing a more decisive parameter when compared to the total FFA concentration according to some authors [57]. Commonly used molar FFA:BSA ratios within the literature were at maximum 5:1. It was suggested to not exceed this ratio [57], whereas due to hyperlipidemia, higher ratios could be suitable for modeling the pathophysiological state in T2DM [58].

The vast amount of the reviewed data (*n* = 199) were obtained from rodent cell culture (rodent *n* = 96, human *n* = 12, monkey *n* = 1) and animal models (Table 4). If comparing the results obtained from cell lines and primary β-cells, there are no remarkable differences if same FFA concentrations are applied [59,60]. There is no indication that there is an adaption of applied FFA concentrations depending on whether rodent cell lines and isolated human islets are used. This leads to the question if FFA concentrations used in rodents are suitable for studying human-based systems.

## 4. Factors Influencing Lipotoxic Outcomes in Animal Models

In animal experiments a high fat diet ranging from 20–60% fat content (*n* = 12) was reported. There was no direct treatment of animals with isolated FFA (*n* = 18).

Animal models (*n* = 73) included 23 studies using wild types (C57BL/6 *n* = 15, C57BL/6J *n* = 8) and 11 using specific mutations of metabolism (C57BL/6 mutants *n* = 8, C57BL/6J mutants *n* = 3), Wistar rats (*n* = 11), Sprague Dawley rats (*n* = 9), db/db (*n* = 4), ob/ob, CD1 mice, Zucker diabetic fatty rats (each *n* = 3) and ICR, NMRI, KK-Ay, Atg7f/f, HcB19 and nu/nu mice (each *n* = 1) (Table 5). In 17 articles, isolated islets from humans were investigated. The mentioned parameters like chosen model, type, and concentration of FFA as well as preparation of FFA stock solutions must be carefully considered while comparing the different results. 

## 5. Lipotoxic Action of FFA on Mitochondria in β-Cells

### 5.1. Detrimental Effects of Elevated FFA in Type 1 and Type 2 Diabetes Mellitus

FFA are well known to mediate toxic effects and impair the function of β-cells enhancing blood glucose levels and increasing the cellular abundance of glucose molecules. Since T2DM patients are exposed to elevated FFA blood levels, they are considered at special risk to suffer from lipotoxic effects including damaged β-cells.

Both reduced insulin levels and action promote lipolysis, but the specific cause why FFA are elevated remain mostly unclear [61]. The abundance of energy generated by the so-called “obesogenic” environment, hallmarked by low physical activity and high caloric, Western-style diet rich in short-chain carbohydrates and animal fats [11], is considered an important link between elevated FFA in T2DM, obesity, and related deteriorations in lipid metabolism. Importantly, a pathological insulin resistance is promoted by lipotoxic effects [43], representing a major mechanism for obesity as a risk factor for T2DM. In type 1 DM, there is no predominant correlation between lipotoxicity and destruction of β-cells [62].

FFA are able to mediate lipotoxicity at different stages of their metabolism covering the range from uptake [63] to degradation [64]. Therefore, lipotoxicity is of a multifactorial etiology and not restricted to a single specific pathway. The following sections will elucidate the underlying pathophysiological mechanisms.

### 5.2. Cellular Uptake of FFA by CD36 and Impairment of Calcium Concentration

FFA uptake into the cell is mediated by fatty acid transporter, also known as cluster of differentiation (CD) 36 [65]. It is known that β-cells abundantly express CD36 [66], probably to ensure a constant FFA supply for energy allocation. CD36 does have substrate specificity, preferring long chain over medium chain FFA [67]. By the binding of long chain FFA to CD36, a pro-inflammatory response is promoted [68] inducing elevated levels of oxidative stress and cellular damage. A relief from oxidative stress can be detected by deletion of CD36 [69]. Another mechanism of FFA uptake is by binding to the free fatty acid receptor 1, also known as the G protein-coupled receptor (GPR) 40 [70]. GPR40 activates phospholipase C (PLC) [71] which is degrading phosphatidylinositol-4,5-bisphosphate into the signaling molecules inositol trisphosphate (IP3) and diacylglycerol to increase the cytosolic calcium (Ca) concentration [72]. Furthermore, Ca storages of the mitochondria are released by PLC, and the endoplasmic reticulum (ER) can release its Ca storages through the activation of GPR40 [71]. The increase of the cytosolic Ca concentration induces exocytosis of insulin vesicles [73]. The replenishment of Ca storages is facilitated by the activation of the sarcoplasmic/endoplasmic reticulum calcium ATPase (SERCA). SERCA activity is stress-sensitive and can be impaired by abundant FFA. As a result, depleted Ca storages render GSIS impossible [74], and Ca-sensitive enzymes of the tricarboxylic acid (TCA) cycle or the transport of NADH as well as the electron transport chain in mitochondria are dysregulated [73].

### 5.3. Mitochondrial Uptake and Processing of FFA

Cytosolic FFA must be activated to acyl-CoA by coenzyme A for further reactions. Acyl-CoA can be transported into the mitochondria by carnitine palmitoyltransferase (CPT) 1, also known as carnitine acyltransferase I [75]. CPT1, the key enzyme of lipid metabolism [76], will exchange the CoA residue with carnitine and initiate the transport across the outer mitochondrial membrane. The transport across the inner mitochondrial membrane is facilitated by carnitine-acylcarnitine translocase and finally, CPT2 will perform the cleavage into carnitine and acyl-CoA in the mitochondrial matrix. Acyl-CoA will be degraded through several enzymatically catalyzed steps of β-oxidation leading to the formation of acetyl-CoA and the reduction equivalents NADH and FADH_2_ [77]. Glycolysis will generate acetyl-CoA and the respective reduction equivalents. Acetyl-CoA from FFA or glucose will be used in the TCA cycle to generate more reduction equivalents. 

An important biochemical feature of lipotoxicity is the impaired activity of glycolytic and TCA cycle-related enzymes with the associated ability for anaplerotic reactions. The impaired enzymes, among others the citrate synthase [78], cause a shortage of intermediates required for the TCA cycle like oxaloacetate, citrate, and α-ketoglutarate [79], which might also be caused by a reduced activity of pyruvate carboxylase [80,81,82], and is also directly proportional to GSIS [83]. The gene expression of those enzymes is impaired [84], and the exchange of pyruvate with TCA cycle intermediates like citrate or malate is abolished and blunts GSIS [85]. 

### 5.4. FFA-Induced Deterioration of Anaplerosis

FFA decrease glutamine levels [86], an amino acid which supports pyruvate transport [87]. Elevated FFA are thought to influence glutamine by increasing the transformation into glutamate [88] and impairing the activity of glutamine synthetase [89]. Decreased glutamine levels further impair the function of β-cells by inhibiting cellular respiration and glucagon-like peptide 1 (GLP-1)-promoted GSIS [86,90] incrementing ROS and the unfolded protein response (UPR).

Interestingly, Lee et al. reported that pyruvate carboxylase inhibition, both by phenylacetic acid (PAA) as well as by high glucose/PA treatment, mitigates AMP-activated protein kinase (AMPK), promoting apoptosis in insulinoma (INS-1) cells. By contrast, AMPK activation was protective against lipotoxic cytotoxicity. PAA and PA treatment promoted CCAAT/enhancer binding protein homologous protein (CHOP) [79] and phosphorylated c-Jun N-terminal kinases (JNK) [91]. The activation of CHOP mediates ATP depletion, consequently reducing GSIS [92]. The reduced flux of metabolites like oxaloacetate, citrate, and α-ketoglutarate worsens the harmful effects of lipotoxicity, whereas the reconstitution of anaplerosis alleviates the consequences of lipotoxicity, as reactions providing intermediates for TCA cycle are enhanced [79]. The authors suggest that the cause for lipotoxicity is closely related to fuel supply by TCA cycle.

In diabetic mouse models it was observed that T2DM correlates with a decreased activity of the pyruvate dehydrogenase complex [93]. This enzyme complex is necessary for glucose utilization and energy production in the TCA cycle. Glucose will be degraded to pyruvate, and through pyruvate dehydrogenase shortened to acetyl-CoA, implicating pyruvate and its related metabolic pathways exert a crucial role for the mitochondrial function [79]. 

### 5.5. Impairment of Iron-Sulfur Cluster Biosynthesis and Ferroptosis Is Induced by FFA

The impaired formation of iron-sulfur (Fe/S) clusters [94,95] was reported to be affected by elevated FFA. These clusters are generated in mitochondria by the iron-sulfur cluster assembly machinery (ISC) [96]. The ISC contains more than 18 proteins responsible for the formation, transfer, or insertion of Fe/S clusters into apoproteins [97]. Fe/S clusters contribute to three-dimensional molecular structure, transfer electrons, or are enzymatic co-substrates. Some of the Fe/S enzymes are involved in energy metabolism like complex I [98], II [99], and III [100] of the respiratory chain. Other Fe/S enzymes serve as sensors for oxygen [101], are involved in gene expression [102] or lipid metabolism [103]. Aconitase, an Fe/S cluster containing enzyme, is especially interesting in this context [104]. It has different functions in the cell depending on its localization. While the mitochondrial aconitase is part of the TCA cycle, the cytosolic aconitase has a regulatory function in Fe-homeostasis and is accordingly termed iron regulatory protein 1 (IRP1) [105]. While losing its Fe/S cluster in an Fe-deficient state, IRP1 binds the iron regulatory element (IRE) of mRNA. IREs are part of the cellular iron-regulatory machinery like ferritin, transferrin receptor 1, divalent metal transporter 1, or ferroportin. By binding IRP1 to IRE, the Fe-uptake into the mitochondria is increased, thus providing more Fe for the Fe/S cluster formation. This regulation by IRP1 is highly conserved and found in yeast, plants, animals, and humans [106]. As previously shown, FFA-induced deficiency of glutaredoxin 5 (Glrx5), a protein of the ISC transferring Fe/S clusters, disrupted the Fe/S cluster insertion into apoproteins [94,95]. FFA-mediated cellular stress or impaired protein maturation could be detrimental for the sensitive clusters and enzymes. Consequently, enzymes of the TCA cycle and complexes of the respiratory chain act less efficiently leading to reduced production of ATP. Furthermore, the Fe-regulation by the cytosolic aconitase is impaired inducing uncontrolled Fe-uptake into the mitochondria. Mitochondrial Fe-overload in combination with elevated ROS promotes the generation of lipid peroxides leading to an Fe-dependent form of non-apoptotic cell death [107] called ferroptosis [60]. Glutathione peroxidase (GPx) 4 is an enzyme which has protective properties by lowering lipid peroxides in a glutathione (GSH) dependent reaction [108] thereby counteracting ferroptosis. Furthermore, FFA also deplete GSH, therefore debilitating the detoxification of lipid peroxides by GPx4 [109]. By increasing oxidative stress, FFA can induce the transformation of GSH into glutathione disulfide [110]. A lipotoxic-induced deficiency of Glrx5 might therefore impair mitochondrial metabolism by lowering activity of TCA cycle enzymes and complexes I-III as well as the induction of β-cell decay by ferroptosis and could be correlated to lower GSIS. Glrx5 mutations have been described in seven human case reports [106,111,112,113,114], of which three describe a link to DM, ferroptosis, and impaired enzyme activities [106,112,114].

### 5.6. FFA Utilization in Energy Metabolism Contributes to Oxidative Stress

In cells with adequate oxygen supply complexes I-IV use reduction equivalents generated by acetyl-CoA to build up a proton gradient in the respiratory chain to generate ATP at complex V (ATP synthase). The electron transfer caused by complex I, complex II, and complex III will generate ROS in a reaction with oxygen [115], complex I contributing to the largest extent [116]. The excessive use of FFA or glucose for energy production will lead to an increased amount of ROS, eventually damaging the cells by reactions with molecules like DNA or enzymes or producing lipid peroxide, triggering ferroptosis. 

Another way of generating ROS out of FFA is the β-oxidation in peroxisomes. Again, it is difficult to define at which specific concentrations ROS become harmful for cell physiology. It is important to note that the imbalance between ROS production and detoxification is the detrimental key factor for oxidative stress. As β-cells have reduced amounts of antioxidative enzymes, it seems likely that the threshold of an overwhelmed antioxidative defense is lower compared to other types of cells. 

There are some data indicating that FFAs mediate an increasing level of toxicity depending on their carbon chain length. Especially long chain fatty acids are suspected to be preferably metabolized in peroxisomes due to specific FFA importers in mitochondria and peroxisomes while less toxic short chain and middle chain FFAs are degraded in the mitochondria [115]. Thus, a diet rich in middle chain fatty acids is not detrimental to β-cells and even promotes GSIS [117], presumably mediated through GPR40.

In human plasma, the five most abundant FFAs, namely OA, PA, stearic, linoleic, and palmitoleic acid, do have a minimum carbon chain length of 16 carbon units and belong to the group of long chain fatty acids. These represent more than 90% of the total amount of FFA in human plasma [118]. The reactions of β-oxidation in peroxisomes are mostly similar to the reactions in the mitochondria, except for the initial step that is facilitated by different enzymes. In peroxisomes the acyl-CoA oxidase will reduce FAD to FADH_2_ and form a double binding in the carbon chain of fatty acids. The acyl-CoA dehydrogenase will perform the same reaction in mitochondria [119]. While the electron transfer to FADH_2_ in mitochondria by acyl-CoA dehydrogenase can be further used for ATP production, in peroxisomes the electrons will interact with oxygen to form hydrogen peroxide (H_2_O_2_). Loading with FFA leads to an increased formation of H_2_O_2_ in peroxisomes as compared to mitochondria [120]. H_2_O_2_ is membrane permeable [121] and can induce negative effects also outside of these organelles and impair insulin secretion. These observations support the hypothesis that ROS production by peroxisomal β-oxidation as well as by mitochondrial respiratory chain complexes are the major reasons for lipotoxicity. Due to a lack of the expression of catalase, GPx1, and superoxide dismutase (SOD), β-cells seem to have only weak protection against oxidative stress [14,122]. Other authors are raising issues that β-cells are also equipped with other antioxidant systems like proteins of the thioredoxin family making them more resistant to stress conditions as generally assumed [123].

### 5.7. Uncoupling and GLP-1 Agonists Relieve Cellular Stress

As an adaptive response to increased oxidative stress, β-cells are able to induce the expression of uncoupling protein (UCP) 2. UCP2 is a proton channel localized at the inner mitochondria membrane dissipating proton gradients [124]. There are four existing isoforms, while only UCP2 is occurring in β-cells [125]. Uncoupling of mitochondria in brown adipose tissue by UCP1 is known as a useful process for generating heat [126]. However, uncoupling in β-cells by UCP2 is not proven as part of thermogenesis [127]. The uncoupling in β-cells is probably a rather adaptive response to increased oxidative stress. The elevation of ROS, especially superoxide as produced by complexes I and III, is required for the upregulation of UCP2 [128,129]. Increased H_2_O_2_ concentrations further activate calcium-independent phospholipase A2 γ (iPLA₂γ) and fuel UCP2 uncoupling [56]. iPLA₂γ alleviates oxidative stress, but it is also a phospholipid remodeling and repair factor of the inner mitochondrial membrane targeting oxidized cardiolipin. Cardiolipin is important for mitochondrial function by regulating gene expression and influencing electron transfer at the respiratory chain complexes [130]. Exendin-4, a GLP-1 agonist used in DM therapy, can increase insulin secretion and reduce β-cell apoptosis by mechanisms including acetylation of iPLA₂γ [131,132] as well as upregulation of pancreatic and duodenal homeobox 1 (Pdx1) also known as insulin promotor factor 1 [133]. The regulation of Pdx1 is additionally mediated by a FFA induced increase of Small heterodimer partner interacting leucine zipper protein, which again increments apoptosis [134]. FFA are ligands for the peroxisome proliferator-activated receptor (PPAR), a receptor responsible for modulation of various pathways in lipid metabolism. There are three subtypes of PPAR, namely PPARα, PPARβ/δ, and PPARγ [135]. All of these subtypes can be activated by FFA and will increase the transcription of UCP2 [4,76,136,137,138,139]. Polymorphisms in the promotor area can also lead to a deterioration of lipid metabolism leading to obesity [140].

### 5.8. ATP Production Is Diminished by Uncoupling and Reduction of ATP Synthase Activity 

The promotor region of UCP2 contains a sterol regulatory element (SRE). Binding by sterol regulatory element binding protein (SREBP)-1c can increase UCP2 expression [141,142]. This process is promoted by FFA. Furthermore, hormone sensitive lipase, an enzyme degrading TG into FFA, activates UCP2 linking elevated FFA levels to UCP2 activation [137].

It was shown that the uncoupling of the proton gradient is dependent on the structure of FFA, and saturated FFA promote uncoupling and exert cytotoxic effects [143]. The uncoupling of the respiratory chain by UCP2 will reduce the amount of ROS, though also lowering the ATP production at complex V. FFA can also influence the ATP synthase regardless of uncoupling factors. The ATP synthase consists of two complexes, F1 and F0. While F0 will decrease the proton gradient, F1 is the catalytic complex forming ATP [144]. The F1 complex is build up by several subunits. Among these, the β-subunit has a crucial role in ATP production because of its ATP binding site. It is reported that FFA can reduce the expression of that specific subunit leading to a decreased ATP production [145]. FFA can also induce acetylation of Sirtuin, which modulates the activity of complex V [146]. 

### 5.9. Membrane Potential Is Modulated by the Abundance of Glucose and FFA, and Impairs Insulin Secretion

The mitochondrial membrane potential (MMP) is dependent on a sufficiently high proton gradient within the mitochondria. FFA are well known to decrease the MMP by an UCP2-induced lowered proton gradient [49]. The reduced MMP is not only linked to reduced ATP amounts and mitochondrial dysfunction, but also to the early stage of apoptosis and therefore β-cell decay [147]. A decreased MMP and a lowered cellular ATP/ADP ratio are counteracting insulin production and secretion of insulin vesicles, connecting mitochondrial function to insulin release. The uncoupling of respiratory chain complexes is a mechanism to relieve cells of oxidative stress, whereas extensive uncoupling subsides the ATP production. As the aerobic ATP production and reduced hyperpolarization of MMP is a crucial factor for inducing insulin secretion [148,149], extensive uncoupling is not favorable for β-cells. If studied in vitro, the lipotoxic effect by uncoupling of the respiratory chain is mostly seen in states of elevated glucose concentrations [59,78]. In a glucose-stimulated state, β-cells are more depending on ATP rendering them more vulnerable against inefficiencies of the respiratory chain activity. Further suggestions indicate, that the lipotoxic uncoupling requires a high membrane potential reached at elevated glucose levels [150]. Interestingly, the impact of different glucose concentrations is suggested to be highly dependent on the employed model. While INS-1 cells show a glucose-dependent increase in apoptosis, no such effect was detectable in mouse insulinoma 6 (MIN6) cells or cultured human islets [151]. In the screened literature, few articles were able to observe such a glucose dependency in other cell lines than INS-1 [59,116,152,153,154,155,156].

### 5.10. PPAR Activity Is Incremented By FFA

Glucose can decrease the action of PPARα leading to limited fatty acid metabolization [138]. This so-called glucose fatty-acid cycle (Randle cycle) ensures a sufficient energy supply to the cells as lipid storages are not used for energy production if enough glucose is available and remain unaffected for phases of starvation. In general, a downregulated rate of fat metabolism with decreased FFA clearance enhances lipotoxic effects by accumulation of lipid intermediates [157] such as acyl-CoA and malonyl-CoA. The accumulation of especially long chain acyl-CoA in the cytosol induces functional impairment and apoptosis by mediating signaling effects and Ca release. Acyl-CoA interacts with PPARα [138] promoting apoptosis [72], uncoupling of respiratory complexes [133], and storage as TG [158], ultimately impairing GSIS [157]. Based on the acyl-CoA content of hamster islet transformed-tioguanine resistant clone 15 (HIT-T15) cells and the average distribution of cytosolic and mitochondrial mass in mouse pancreas [159], it is estimated that the cytosolic acyl-CoA concentration in rodent β-cells is 90 µM, while tissue concentrations are unknown [160]. Malonyl-CoA inhibits CPT1 preventing the mitochondrial uptake of acyl-CoA [156]. Yet, there are contrary conclusions regarding the influence of CPT1 in the context of lipotoxicity. While two publications indicate that mitochondrial FFA oxidation is required for lipotoxic effects and can be counteracted by the inhibition of CPT1 [93,138], another study suggests that the inhibition of CPT1 has no effect on lipotoxicity [57] indicating that FFA mediate lipotoxicity independently from mitochondrial metabolism. 

### 5.11. The Process of Autophagy Is Disturbed by FFA

PPARγ regulates the gene expression responsible for autophagy [139] and apoptosis. While low grade autophagy activity is a requirement for remodeling of damaged cellular components [161], enhanced autophagy leads to disintegration of the β-cell [162]. In case of mitochondria, this process is called mitophagy. In the first step, damaged cellular components form autophagosomes [43], which are degraded by lysosomes. The activity of lysosomes is depending on ATP needed for acidification [43]. Thereby, FFA can reduce the lysosomal activity [163,164]. This will lead to an accumulation of autophagosomes [161]. Additionally, FFAs can disturb the process of autophagy by lowering the expression of the mechanistic target of rapamycin (mTOR) [43] and overexpression of optic atrophy protein 1 [163] or dynamin-related protein 1 [165]. An increase in mTOR by FFAs is also related to insulin resistance [43].

### 5.12. Acyl-CoA Abates Insulin Synthesis in Β-Cells

Acyl-CoA is able to degrade the proton gradient of the respiratory chain through the formation of mitochondrial permeability transition pore [166] disturbing GSIS [71]. An increase in long chain acyl-CoA can inhibit the closure of ATP-dependent potassium (K)-channels [167,168]. Factors involved in this effect are the acyl group, the CoA component, and protein kinase C [169]. FFA can open K-channels by direct interaction [170] or as a consequence of GPR40 mediated Ca influx, which is both decreasing the ability of glucose to stimulate insulin secretion. In addition, the mRNA level of insulin is reduced by acyl-CoA [156] and palmitoylation [57,171], as well as the interaction with hepatic nuclear factor 4a, which could be related to uncoupling by UCP2 [45]. Some authors claim that the activation of FFA into acyl-CoA is one of the most essential aspects mediating harmful effects as the inhibition of acyl-CoA synthase could suppress lipotoxicity-mediated cell death [172,173]. 

### 5.13. Ceramides Increase Oxidative Stress through Inducible Nitric Oxide Synthase

Palmitoyl-CoA, the activated form of PA, is substrate for the *de novo* formation of ceramides and upregulates sphingosine kinase 2 (SK2), a key enzyme of ceramide synthesis. Ceramides have impact on various pathways including proliferation, differentiation, growth arrest, and apoptosis [174,175]. They can impair insulin sensitivity by protein kinase C [138] and phosphorylation of insulin receptor substrate 1 (IRS-1) [176] as well as reduce insulin expression by Pdx1 [78]. IRS-1, which is likewise regulated by SREBP-1c [153], is integrated in proliferation signals by PI3K and Akt, while Akt also regulates glucose uptake by GLUT4. Furthermore, ceramides disrupt the acetylation of proteins of the mitochondrial metabolism [177], inhibit complex III, and decrease MMP [42]. They increase cellular oxidative stress through the activity of inducible nitric oxide synthase (iNOS) producing nitrogen oxide [57,79,139,178,179] and NADPH oxidase 2 (NOX2) producing superoxide [55,178,180,181,182]. The activation of both iNOS and NOX2 induces apoptosis by damaging mitochondrial DNA [155], which is more susceptible to harm due to the absence of introns [179]. Damaged DNA can be recognized by stimulator of interferon genes (STING), increasing inflammation and apoptosis. The STING pathway is enhanced under lipotoxic conditions and activates interferon regulatory factor 3 [47]. It has been shown that the presence of ceramides is crucial for the induction of apoptosis. While the regulation of mitogen-activated protein kinase (MAPK) pathway including JNK and extracellular-signal-regulated kinase (ERK) as well as Pi3K/Akt pathway are undoubtedly essential for cell survival, there are contrasting data regarding their exact role for the β-cell. According to literature, effects differ crucially between cell types and highly depend on the chosen treatment conditions [132,183], e.g., PA inhibited ERK and induced apoptosis in INS-1 cells [184], but exposure of glucose and interleukin (IL)-1β triggered apoptosis concomitant with elevated ERK levels in human β-cells [44,185].

Additionally, SK2 was reported to increase apoptosis through Bcl-2, while SK2 inhibition prevented lipotoxic cell death [186]. The regulation of apoptosis by ceramides is seen as one of the most important factors for β-cell decay mediated by lipotoxicity [187]. While unsaturated FFA are not able to increase ceramides [188], other data indicate facilitated ceramide production through an increased ω6:ω3 ratio of fatty acids [189].

### 5.14. Augmented Apoptosis in Β-Cells by Long-Chain and Saturated FFAs

The initiation and execution of the mitochondrial apoptosis pathway is complex and highly regulated. Briefly, the membrane of damaged cells starts to permeabilize, which is enhanced by Bax or p53-upregulated modulator of apoptosis and suppressed by Bcl-2 [190]. The permeabilization is promoted by several stress markers related to the ER, like CHOP, or the activating transcription factor [191]. The release of pro-apoptotic factors eventually triggers downstream caspases like caspase 3. In consequence, chromatin will condensate, DNA will be fragmented, and apoptosis is executed. Particularly long chain and saturated FFA as opposed to intermediate chain length and unsaturated FFAs induce ER stress [64] and counteract protective factors such as Sirtuin3 [44,91]. The fragmentation of DNA is further promoted by oxidative stress sensitive transient receptor potential melastatin-2 channels, inducible by lipotoxicity [55]. 

## 6. Positive Effects of FFA on β-Cell Function

The number of articles reporting beneficial effects of FFA to β-cells and insulin secretion is remarkably low. Most intriguingly, they share the same underlying pathways as detrimental effects. Although this seems to pose a contradiction at first, a more detailed look into the respective methodologies can give possible explanations.

FFA concentrations employed in studies delivering evidence for a beneficial impact were notably lower [71,170,171,192]. While publications reporting detrimental effects of FFA usually applied concentrations around 500 µM, favorable effects were reported with concentrations far below 100 µM. Furthermore, shorter treatment times were associated with positive results indicating a difference between the acute and chronic exposure to FFA. Acute effects were observed after a few hours of treatment while chronic effects were noticeable after at least 24 h. Exposure of INS-1 cells to 100 µM for several weeks severely blunted their insulin secretion (−80%) [49], whereas it was promoted by 1 h incubation with 150 µM PA [56]. The short time treatment with low FFA concentration led to the activation of PPARγ coactivator 1α/β [193] and a depletion of Ca storages by GPR40, mediating insulin release [170]. The so-called “fatty acid stimulated insulin secretion” (FASIS) is induced by acute FFA uptake and oxidation through the activation of cellular energy production [72,158,194]. Resulting physiological low concentrations of ROS are also promoting insulin secretion, which is referred to as “redox stimulated insulin secretion” (RSIS) [56]. This effect is explained by PTEN-induced kinase 1 (PINK1)-mediated autophagy activation, which is improving net insulin release pattern by disposal of damaged cells [162]. In contrast, chronic elevation of FFA exerts a detrimental impact on insulin secretion through permanent depletion of Ca-storages, and an abundance of ROS. 

Another decisive aspect in lipotoxicity is the homeostasis between FFA and fat storage mobilization. The available data indicate that stored neutral TG do not act lipotoxic in contrast to FFA [195]. Therefore, it is crucial to differentiate between an energy surplus leading to an excess of FFA on one hand and a storage as neutral TG on the other hand. While adipocytes have a nearly unlimited TG storing capacity, fat storing in β-cells is limited [195,196]. There are some articles implying that unsaturated FFA promote TG storage, thereby counteracting the toxic effect of saturated FFA [195,197]. Other authors assume that the protective effect is independent from fat storage [198], but rather reliant on a positive effect of unsaturated FFA to proapoptotic factors [46], while also inducing mitochondrial apoptosis [199]. Further data suggest an anti-lipotoxic effect of unsaturated FFA [200]. However, the underlying mechanisms have not yet been elucidated. Unsaturated FFA reveal their anti-lipotoxic effect mainly in co-treatment with saturated FFA. As sole treatment with unsaturated FFA is also toxic, enhanced lipid storing by unsaturated FFA should be further investigated as protective mechanism. It has been shown that the amount of unbound FFA is a better parameter for lipotoxicity instead of total concentration of FFA [57]. A summary of beneficial effects to β-cell physiology can be found in Table 6.

## 7. Influence of Ageing on β-Cell Function

Since there are multiple mechanisms by which lipotoxicity can impair β-cell function, this section will review which of the underlying pathways are especially related to accelerated ageing and senescence of β-cells and the pancreas. Cellular senescence is defined by a cell cycle arrest triggered by damaged DNA, exerting an anti-tumorigenic purpose [201]. It can be induced due to telomere shortening at the end of a cell’s life span, or, dependent on the physiological conditions, as part of the cellular stress response. Lipotoxicity is linked to cellular senescence by the p38 MAPK pathway which is age-dependently correlating with decreased cell proliferation and insulin release in β-cells. Lipotoxicity can contribute to the activation of p38 MAPK by elevated ROS and ceramides. p38 MAPK promotes senescence through different pathways, which are dependent on and independent of telomerase length [202]. As autophagy can be regulated by MAPK, it is tempting to speculate that lipotoxicity might thereby impair the degradation of damaged organelles and cellular regeneration. Age-dependently, ROS and levels of radicals tend to accumulate and lead to enhanced damaging of mitochondrial proteins and DNA, leading to cellular dysfunction. Beside cellular damaging and signal pathways, glucose and insulin homeostasis is impaired by decline in mitochondrial ATP synthesis capacity and reduced expression and translocation of GLUT2 [117]. In addition, depending on age, the insulin dependent uptake by GLUT4 is decreased by Ras-related C3 botulinum toxin substrate 1 (Rac1), a protein which is correlating with ceramide-mediated senescence [203]. In presence of ceramides, the activation of Rac1 could also influence oxidative stress by induction of NOX2 [178]. With increasing age, the activity of insulin like growth factor (IGF) binding protein 3 as well as the total amount of IGF decline. A reduced binding of IGF leads to the deterioration of glucose tolerance, lipid metabolism, and increased stress by absent UCP regulation [204]. FFA can further have negative effects on the activity of the subunits of farnesyl transferase (FTase) and geranylgeranyl transferase (GGTase). FTase and GGTase are enzyme complexes required for the prenylation of proteins determining localization and transport of proteins. The degradation of subunits of FTase and GGTase by FFA activation of caspase is chronically activating Rac1 leading to increased cellular stress. The impairment of protein prenylation has negative consequences on GSIS and is connected to ageing syndromes [180]. A further increase of senescence has been seen in involving caveolin-1, a membrane protein inducing apoptosis by Src family kinases-mediated phosphorylation of tyrosine-14, which is promoted by FFA [42]. While they are known ways, how lipotoxicity is correlating with acceleration of β-cell senescence, the data in this literature search are limited to few articles (*n* = 15).

## 8. Potential Protective Effects of Plant-Based Nutrients

A healthy lifestyle involving sufficient exercise and a balanced diet is a pivotal part of the treatment of T2DM. Plant-based nutrients as mainly recommended in the so-called Mediterranean diet contain numerous phytochemicals mediating wholesome effects. In particular, the major group of polyphenols was reported to wield protective properties, e.g., ameliorating glucotoxicity [205,206], oxidative stress [207], and ER stress [208] as well as inhibiting α-amylase [209,210] and preventing protein glycation [211]. Yet, their effects on the mitochondria of β-cells have been barely investigated (in cell lines: *n* = 5, in animal experiments: *n* = 5). Most interestingly, the beneficial effects of polyphenols are targeting pathways affected by lipotoxicity. They counteract the dysregulation of the MAPK and Pi3K pathway [22,23,212,213,214], and mitigate apoptosis by altering Bax/Bcl ratio and up-regulation of Pdx1, protein kinase A, and cAMP response element-binding protein signaling [215]. They stabilize the mitochondrial membrane preventing the release of proapoptotic factors and enhancing Ca signaling [23]. Moreover, polyphenols support insulin release through interaction with Pdx1 [212] in a forkhead box protein O1 (FoxO1)-dependent manner [214] and inhibit dipeptidylpeptidase-4 (DPP4) to increase GLP-1 levels–a pathway successfully used in clinical practice by DPP4 inhibitors and GLP-1 receptor agonists [33,216,217]. In addition, the dysregulation of GLUT2 and GLUT4 can be reversed by polyphenols improving elevated blood glucose levels and increasing insulin sensitivity [22]. Furthermore, relief from oxidative stress due to their radical scavenging capacity [33], and amelioration of ROS production by iNOS was reported. Polyphenols promote the activity and regeneration of antioxidative enzymes like SOD, catalase, GSH reductase, and GPx [21,33,212,218]. Particularly, GPx4 reduces lipid peroxides in the context of ferroptosis. As lipotoxicity-induced ferroptosis is also closely related to restrictions of Fe/S enzymes, plant-based nutrients rich in polyphenols were shown to sustain the activity of those enzymes, e.g., cytochrome C oxidase or succinate dehydrogenase, which showed lowered activity in an animal study with high fat diet [212] counteracting mitochondrial dysfunction. Possible beneficial effects of polyphenols could be also mediated by improved fat metabolism through upregulation of SREBP-1c including signals from Akt or the estrogen receptor α [22,23] and lipid storing, improving lipid parameters [21,218]. Polyphenols further exert anti-inflammatory effects through decreased cytokine levels and ER stress [212,219].

According to these data, plant-based nutrients rich in polyphenols might be promising agents to counteract lipotoxic damage to β-cell mitochondria in diabetes and account for the beneficial effects of the Mediterranean diet. Yet, their clinical and therapeutical benefit has not been studied in clinical trials, and some authors even claim that toxic effects might arise from dosages required to achieve therapeutic levels in humans [33]. A summary of the described effects is given in Table 7.

## 9. Conclusions

Mitochondrial function is a key parameter crucially determining energy supply and cell survival. By those factors, it plays a central role in β-cell decay and development of T2DM. FFAs are well known to impair the glucose metabolism by mediating negative effects on mitochondria generally known as lipotoxicity. As lipotoxicity is multifactorial, most of the reviewed studies described lipotoxic effects mediated by increased oxidative stress, uncoupling of energy production by UCP2, deterioration of lipid homeostasis by PPAR and SREBP-1c signaling, extra-mitochondrial signaling through accumulation of acyl-CoA, and eventually enhanced apoptosis. While lipotoxicity can accelerate senescence of β-cells, there is evidence for a sustained mitochondrial metabolism and reversed effects of lipotoxicity by polyphenols as apparent in plant-based nutrients. However, the data are extremely limited and covering a wide range of different plants and ingredients, preventing a distinct verdict on their significance for T2DM and lipotoxicity. When studying the effects of FFAs, parameters like structure, concentration, treatment duration, and preparation should be considered carefully since they have massive impact on the outcome of the experimental setup. Especially the employed concentration substantially determines if FFAs will have an adverse or beneficial effect on β-cells. Judging cautiously from literature it can be generally assumed that in vitro FFAs mediate physiological effects in the lower micromolar range (Table 6), whereas concentrations in the upper micromolar range and higher exert lipotoxic effects. In contrast, concentrations are considerably higher in vivo with human serum containing levels in the lower millimolar range and murine serum with approximately 10% of this amount.

The different results of FFAs and polyphenol treatment as well as the multifactorial presentation of lipotoxicity leaves several questions unanswered. Therefore, further studies with clearly defined experimental setups would benefit this promising field of research and further elucidate the execution of lipotoxicity and respective protective mechanisms for the pancreatic β-cell.

## Figures and Tables

**Figure 1 antioxidants-10-00293-f001:**
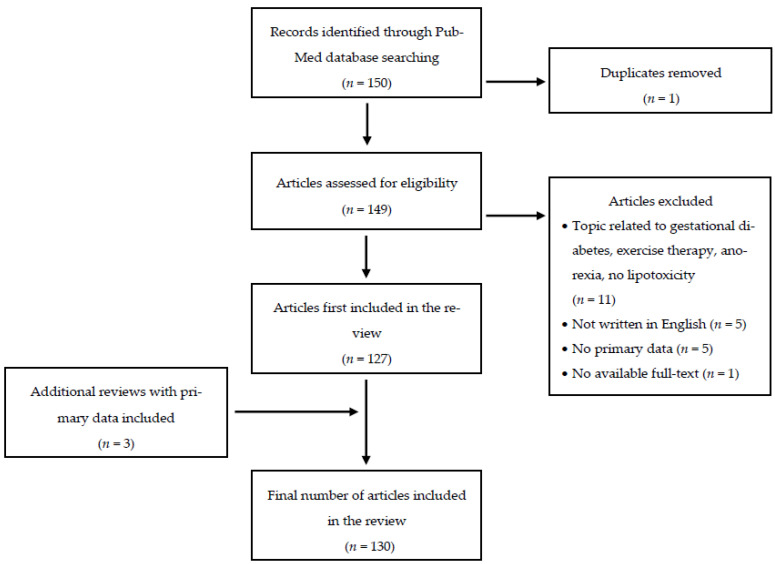
Flow diagram of the literature search on PubMed, www.ncbi.nlm.nih.gov/pubmed.

**Table 1 antioxidants-10-00293-t001:** Frequency of FFA concentrations used in screened literature focused on lipotoxicity (*n* = 132). Free fatty acids (FFA).

Concentration of FFA	Frequency in Screened Articles
10–99 µM	12
100–499 µM	53
500 µM	49
501–2000 µM	18

**Table 2 antioxidants-10-00293-t002:** Frequency of FFA and FFA combinations used in screened literature (*n* = 96). Free fatty acids (FFA). Palmitic acid (PA). Oleic acid (OA). Any FFA is referring to myristic, stearic, palmitoleic, linoleic, linolenic, methylpalmitic, docosahexaenoic, and arachidonic acid.

FFA	Frequency in Screened Articles
PA	52
PA and OA	20
PA and OA and any FFA	8
PA and any FFA	6
OA	6
Any FFA	4

**Table 3 antioxidants-10-00293-t003:** Frequency of used solvents in screened literature (*n* = 98).

Solvent	Frequency in Screened Articles
Ethanol	25
NaOH or NaCl	12
Dimethyl sulfoxide	4
Methanol	3
No further information	54

**Table 4 antioxidants-10-00293-t004:** Frequency of cell culture models used in screened literature (*n* = 109). Insulinoma (INS-1). Mouse insulinoma 6 (MIN6). Human embryonic kidney 293 (HEK 293). Rat insulinoma (Rinm5f). Hamster islet transformed-tioguanine resistant clone 15 (HIT-T15). Hepatoblastoma (HepG2). Chinese hamster ovary (CHO). NOD/Lt (NIT1). β-tumour cell (βTC6). CV-1 in origin simain-1 (COS1).

Cell Line	Frequency in Screened Articles
INS-1 (rodent)	57
MIN6 (rodent)	23
HEK 293 (human)	7
Rinm5f (rodent)	5
HIT-T15 (rodent)	5
HepG2 (human)	3
CHO (rodent)	2
EndoC-βH1 (human)	2
NIT1 (rodent)	2
βTC6 (rodent)	1
BRIN-BD11 (rodent)	1
COS1 (monkey)	1

**Table 5 antioxidants-10-00293-t005:** Frequency of animal models used in screened literature (*n* = 73). Institute for Cancer Research (ICR). Naval Medical Research Institute (NMRI). Diabetic KK and lethal yellow (Ay) mice (KK-Ay). Ubiquitin-like modifier-activating enzyme (ATG7). TXNIP deficiency (HcB19).

Animal Model	Frequency in Screened Articles
wild type C57BL/6 and C57BL/6J mouse	15 and 8
mutant C57BL/6 and C57BL/6J mouse	8 and 3
Wistar rat	11
Sprague Dawley rat	9
db/db mouse	4
ob/ob mouse	3
CD1 mouse	3
Zucker diabetic fatty rat	3
ICR mouse	1
NMRI mouse	1
KK-Ay mouse	1
Atg7f/f mouse	1
HcB19 mouse	1
nu/nu mouse	1

**Table 6 antioxidants-10-00293-t006:** Summary of metabolic effects of FFA treatment and respective pathways. Palmitic acid (PA). Insulinoma (INS-1). Glucose-stimulated insulin secretion (GSIS). Adenosine triphosphate (ATP). Calcium (Ca). Free fatty acids (FFA). Hamster islet transformed-tioguanine resistant clone 15 (HIT-T15). Phospholipase C (PLC). Endoplasmic reticulum (ER). Peroxisome proliferator-activated receptor (PPAR). Institute for Cancer Research (ICR). Chinese hamster ovary (CHO). Human embryonic kidney 293 (HEK 293). Hepatoblastoma (HepG2). Mouse insulinoma 6 (MIN6). Uncoupling protein (UCP). G protein-coupled receptor (GPR). PTEN-induced kinase 1 (PINK1). Oleic acid (OA). Triglyceride (TG).

Article	Treatment	Model	Results and Respective Pathways
Green et al., 2009 [192]	50 µM PA, 1 h	INS-1 cells, human islets	- liver X receptor improved GSIS - β-oxidation provided ATP for GSIS - lipid signaling supported Ca influx
Komatsu et al., 1999 [171]	10 µM PA, 1 h	Wistar Rat islets	- FFA supported GSIS within first 10 min of secretion
Remizov et al., 2003 [170]	100 µM PA, 30-60 min	HIT-T15 cells, primary mice β-cells	- FFA caused Ca mobilization from internal storages
Zhao et al., 2013 [71]	20 µM linoleic acid, 2-10 min	Sprague Dawley rat islets	- FFA stimulated Ca increase. Effect depended on Acyl-CoA synthase, PLC, and ER/mitochondrial Ca storages
Oropeza et al., 2015 [193]	100 µM PA, 1 h	C57BL/6J mice islets	- FFA increased PPARγ coactivator 1α expression, regulating key enzymes in lipolysis and the glycerolipid/free fatty acid cycle
Chen et al., 2020 [72]	10 µM linolenic acid, 1 h	INS-1 cells, KO mice islets, Wistar Rat islets	- FFA receptor 1 agonist supported insulin secretion by increased mitochondrial function and β-oxidation
Li et al., 2020 [194]	10 µM linolenic acid, 1 h	ob/ob mice, ICR mice, C57BL/6 mice, CHO cells, HEK293 cells, HepG2 cells, MIN6 cells	- FFA receptor 1 agonist supported insulin secretion and glycemic control
Li et al., 2020 [158]	No FFA	C57BL/6 mice, ob/ob mice, db/db mice	- FFA receptor 1/PPAR agonist supported β-cell function and fatty acid metabolism
Ježek et al., 2015 [56]	150 µM PA, 1 h	INS-1 cells	- FFA activated UCP2. Oxidative stress by physiological FFA uptake was prevented. - PA increased insulin secretion by GPR40
Guo et al., 2019 [162]	100–500 µM PA, 24–48 h	RIN-m5f cells	- sonodynamic therapy increased insulin secretion of damaged cells by activated PINK1 autophagy
Cho et al., 2012 [195]	100–500 µM PA, 24 h and 10–62 µM arachidonic acid and 20–120 µM unsaturated FFA (OA, arachidonic acid, palmitoleic acid)	HIT-T15 cells	- unsaturated fatty acids protected against PA damages, probably by TG accumulation
Tuo et al., 2011 [199]	50–500 μM linoleic acid, 48 h	INS-1 cells	- negative effects occurred from 250 µM upwards for viability, effect depended on high glucose concentrations
Ježek et al., 2018 [200]	100 μM PA, 10-60 min	C57BL6J mice islets	- monoacylglycerol bound to GPR119 and enhanced insulin secretion
Cnop et al., 2001 [57]	125–500 μM PA and OA, 2 d and 8 d	Wistar Rat islets	- OA treatment accumulated more TG than PA, ameliorating the detrimental effects of FFA

**Table 7 antioxidants-10-00293-t007:** Summary of metabolic effects mediated by plant-based polyphenols and respective pathways. Oleic acid (OA). Mouse insulinoma 6 (MIN6). Glutathione peroxidase (GPx). Free fatty acids (FFA). Glucagon-like peptide 1 (GLP-1). Dipeptidylpeptidase-4 (DPP4). Reactive oxygen species (ROS). Superoxide dismutase (SOD). Triglyceride (TG). Palmitic acid (PA). Insulinoma (INS-1). Pancreatic and duodenal homebox 1 (Pdx1). Extracellular-signal-regulated kinase (ERK). Glucose-stimulated insulin secretion (GSIS). Endoplasmic reticulum (ER). Insulin receptor substrate 1 (IRS-1). Glucose transporter (GLUT). Rat insulinoma (Rinm5f). AMP-activated protein kinase (AMPK). Mechanistic target of rapamycin (mTOR). High fat diet (HFD). Interleukin (IL). Tumor necrosis factor α (TNFα). Forkhead box protein O1 (FoxO1).

Article	Treatment	Model	Extract, Substance	Results and Respective Pathways
Zakłos-Szyda et al., 2020 [33]	100 µM OA, 24 h	MIN6 cells	*Viburnum opulus* L., fresh juice and phenolic rich fraction with chlorogenic acid, flavanols, procyanidins	- reduced oxidative stress by radical scavenging and activation of antioxidative enzymes (GPx) - increased FFA uptake and lipid droplets accumulation - increased GLP-1 secretion by inhibited DPP4 activity - higher extract dosages increased necrosis/apoptosis by caspase activation and elevated ROS
Renganathan et al., 2020 [21]	No induction	Wistar rats	Dhanwantaram kashayam, polyherbal formulation containing *Sida spinosa* L., *Hordeum vulgare* L., *Aegle marmelos* (L.) Corrêa, *Bauhinia forficata* Link.	- less oxidative stress by activation of antioxidative enzymes (catalase, SOD, GPx, glutathion reductase) - improved lipid parameters (total cholesterol, FFA, phospholipids, TG) - extract abated antioxidative enzymes in control rats
Liu et al., 2019 [34]	200 µM PA, 24–96 h	INS-1 cells, C57BL/6J mice islets	Dracorhodin perchlorate	- increased Pdx1 expression by ERK1/2 - decreased apoptosis by Bax/Bcl-2 ratio - lowered blood glucose by improved GSIS, increased islet size/number - diminished ER stress
Sun et al., 2019 [23]	100 µM PA, 48 h	INS-1 cells	Silibinin	- improved viability, GSIS, lipid metabolism by estrogen receptor - increased mitochondrial mass and improved mitochondrial membrane potential
Gharib and Montasser Kouhsari, 2019 [22]	No induction	Wistar rats	*Punica granatum* L., fruit extract with punicalagin, anthocyanins, ellagic acid, gallic acid, caffeic acid, catechins, quercetin, rutin	- lowered fasting glucose by modulations of IRS-1, Akt, GLUT2/4 mRNA - enhanced lipid parameters (FFA, TG)
Gharib et al., 2018 [213]	No induction	Wistar rats	*Punica granatum* L., fruit extract with punicalagin, anthocyanins, ellagic acid, gallic acid, caffeic acid, catechins, quercetin, rutin	- improved lipid parameters (FFA, TG) - increased insulin sensitivity by decreased p53, p65, miR-145 and elevated IRS-1 - reduced ROS
Huang et al., 2017 [216]	100 µM PA, 24 h	RINm5F cells	*Abelmoschus esculentus* (L.) Moench, extract with quercetin glucosides, pentacyclic triterpene ester, carbohydrates, polysaccharides	- increased GLP-1 effect by decreased DPP4 activity - reduced apoptosis by AMPK, mTOR, PI3K signaling
Liu et al., 2017 [212]	HFD, 6 weeks	Sprague dawley rats islets	*Morus nigra* L., leaf extract with polysaccharides	- improved lipid parameters (FFA, TG, low-density lipoprotein) - decreased IL-6, TNFα - lowered fasting glucose - promoted mitochondrial enzymes (succinate dehydrogenase, cytochrome C oxidase) - morphological improvement of β-cells
Hao et al., 2015 [214]	500 µM PA, 24 h	MIN6 cells	Curcumin	- reduced apoptosis by caspase and Bax/Bcl-2 ratio - improved GSIS by mitochondrial membrane potential, Akt, FoxO1 - reduced oxidative stress by antioxidative enzymes (MnSOD, catalase, GPx, glutathione reductase)

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
