# Peer review of "Lipotoxic Impairment of Mitochondrial Function in β-Cells: A Review"

_antioxidants, 2021, doi:10.3390/antiox10020293_

Round 1

Reviewer 1 Report

Comments to the Author

The authors summarized the significance of lipotoxicity and cellular senescence for mitochondrial dysfunction in the pancreatic β-cell, and how plant extracts might exert beneficial effects in diabetes. However, there are some concerns in the manuscript that once corrected will improve the quality of this work.

Major criticisms

  1. In Table 1, the authors described that screening of the available literature revealed significant differences in the employed concentrations of FFA. However, the relation between the effects (or effective degree) and different concentrations of FFA should be included.
  2. In Table 2. the authors indicated that frequency of FFA and FFA combinations used in screened literature (n = 94). The combined doses of PA and OA, PA and OA and any FFA, and PA and any FFA should be provided.
  3. Possible beneficial effects of polyphenols against lipotoxicity and cellular senescence, maintaining the physiology of β-cells should be provided in detailed table.
  4. In contrast with various cellular or animal models (Table 4 or 5), the apparent contradictory data in the literature in different models may reflect differences in experimental design, specifically in the concentration and time of induction. It should further be discussed.
  5. The authors should take care of numerous grammatical mistakes throughout the manuscript which detract from its quality.

Reviewer 2 Report

In the present manuscript, the authors aimed to summarize the mechanisms of lipotoxicity in diabetes mellitus and the potential beneficial effect of plant extracts.

It was disappointing that only 1 page out of 15 was actually about the effects of plant extracts without any introduction to these substances and any categorization of their possible effects. Talking about hundreds of different chemical agents without differentiation of their characteristics is misleading and sweeping generalization. Vitamin E is not a plant extract, and earching for clinical trials with Viamin E in diabetes would show tens of published clinical trials.

The manuscript was many other weaknesses. It contains several general statements and listings without proper definitions or lacking novel information. Like in the second paragraph: “ dysregulation and enhancement of stress” (What kind of stress?).

Not including flavonoids in the searching procedure may also had impact on the composition of the manuscript.

After the collection of relevant articles about the topic, the possible mechanisms of lipotoxicity is discussed for 9 pages. Although it could be a relevant and important summary, it should not be the main chapter in a review that tries to summarize the effects of plant extracts. This part of the article is not well structured, contains several repetitions and some erroneous statements too.

Examples:

I do not understand, how the inhibition of pyruvate carboxylase (an enzyme of gluconeogenesis) can lead to AMPK activation. I found no connection in the cited references either. (In ref 55, there is no information about PC at all, checking the previous ones 76-78, I still did not find any mentions of this phenomenon.

GPR40 is a Gq coupled receptor. It increases intracellular Ca++ level and induce insulin release. Its signal transduction does not include opening of potassium channels as stated in the manuscript. The Ca dependent channels were not even specified to be K+ channels. Indirectly by the increased Ca++ level, K+ channel may open, but this is not a direct effect. KATP channels will open due to reduced ATP/ADP ratio if the detrimental effects of FFAs lead to decreased mitochondrial function. There are other GPRs binding FFAs that can open K+ channels too, but it is not the GPR40.

MAPK, most precisely ERK pathway is not pro-apoptotic, it is pro-proliferation and pro-survival pathway. It should not be forgotten that the role of p38 MAPK role is senescence is in strong correlation with its anti-apoptotic properties.

Reviewer 3 Report

The manuscript is well written and is recommended for publication.

Round 2

Reviewer 1 Report

Comments to the Author

The authors summarized the significance of lipotoxicity and cellular senescence for mitochondrial dysfunction in the pancreatic β-cell, and how plant extracts might exert beneficial effects in diabetes. However, there are some concerns in the manuscript that once corrected will improve the quality of this work.

Major criticisms

  1. In Table 1, the authors described that screening of the available literature revealed significant differences in the employed concentrations of FFA. However, the relation between the effects (or effective degree) and different concentrations of FFA should be included.
  2. In Table 2. the authors indicated that frequency of FFA and FFA combinations used in screened literature (n = 94). The combined doses of PA and OA, PA and OA and any FFA, and PA and any FFA should be
  3. Possible beneficial effects of polyphenols against lipotoxicity and cellular senescence, maintaining the physiology of β-cells should be provided in detailed table.
  4. In contrast with various cellular or animal models (Table 4 or 5), the apparent contradictory data in the literature in different models may reflect differences in experimental design, specifically in the concentration and time of induction. It should further be discussed.
  5. The authors should take care of numerous grammatical mistakes throughout the manuscript which detract from its quality.

Reviewer 2 Report

The manuscript has been substantially improved.

The two articles with the searching term flavonoids is not involved in the general description of used articles in section 2 and fig. 1.

According to this figure, the authors involved 124 articles in the review. However they have 219 references. They should indicate which references are part of the systematic search, and which are just additional references. 
